# Regulation of Glucose Metabolism by MuRF1 and Treatment of Myopathy in Diabetic Mice with Small Molecules Targeting MuRF1

**DOI:** 10.3390/ijms22042225

**Published:** 2021-02-23

**Authors:** Siegfried Labeit, Stephanie Hirner, Julijus Bogomolovas, André Cruz, Moldir Myrzabekova, Anselmo Moriscot, Thomas Scott Bowen, Volker Adams

**Affiliations:** 1Department of Anesthesiology, Medical Faculty Mannheim, University of Heidelberg, 68169 Mannheim, Germany; stephanie.hirner@gmx.de; 2Myomedix GmbH, 69151 Neckargemünd, Germany; 3Zentralinstitut für Seelische Gesundheit, 68159 Mannheim, Germany; jbogomolovas@ucsd.edu; 4Department of Anatomy, Institute of Biomedical Sciences, University of Sao Paulo, 05508-000 Sao Paulo, Brazil; andrecruz@usp.br (A.C.); moriscot@usp.br (A.M.); 5Scientific Research Institute of Biology and Biotechnology Problems, al-Farabi Kasakh National University, Almaty 050040, Kazakhstan; moldir.myrzabek@gmail.com; 6School of Biomedical Sciences, University of Leeds, Leeds LS2 9JT, UK; T.S.Bowen@leeds.ac.uk; 7Laboratory of Molecular and Experimental Cardiology, TU Dresden, Heart Center Dresden, 01307 Dresden, Germany; volker.adams@mailbox.tu-dresden.de; 8Dresden Cardiovascular Research Institute and Core Laboratories GmbH, 01307 Dresden, Germany

**Keywords:** diabetes mellitus, glucose and muscle metabolism, MuRF1, MuRF2, chemical biology

## Abstract

The muscle-specific ubiquitin ligase MuRF1 regulates muscle catabolism during chronic wasting states, although its roles in general metabolism are less-studied. Here, we metabolically profiled MuRF1-deficient knockout mice. We also included knockout mice for MuRF2 as its closely related gene homolog. MuRF1 and MuRF2-KO (knockout) mice have elevated serum glucose, elevated triglycerides, and reduced glucose tolerance. In addition, MuRF2-KO mice have a reduced tolerance to a fat-rich diet. Western blot and enzymatic studies on MuRF1-KO skeletal muscle showed perturbed FoxO-Akt signaling, elevated Akt-Ser-473 activation, and downregulated oxidative mitochondrial metabolism, indicating potential mechanisms for MuRF1,2-dependent glucose and fat metabolism regulation. Consistent with this, the adenoviral re-expression of MuRF1 in KO mice normalized Akt-Ser-473, serum glucose, and triglycerides. Finally, we tested the MuRF1/2 inhibitors MyoMed-205 and MyoMed-946 in a mouse model for type 2 diabetes mellitus (T2DM). After 28 days of treatment, T2DM mice developed progressive muscle weakness detected by wire hang tests, but this was attenuated by the MyoMed-205 treatment. While MyoMed-205 and MyoMed-946 had no significant effects on serum glucose, they did normalize the lymphocyte–granulocyte counts in diabetic sera as indicators of the immune response. Thus, small molecules directed to MuRF1 may be useful in attenuating skeletal muscle strength loss in T2DM conditions.

## 1. Introduction

A secondary progressive loss of muscle mass or sarcopenia in chronic diseases is an increasing medical issue, particularly in the ageing population of developed countries [1,2]. The underlying causes are frequently multifactorial, such as aging-dependent inactivity and malnutrition. Myofibril loss, also referred to as sarcopenia, becomes further exacerbated when patients suffer from additional cachexia-promoting states, such as diabetes, sepsis, or inflammation, that further promote general muscle catabolism [3,4]. The molecular mechanisms that underlie the development of muscle wasting include downregulation of the translation of muscle proteins, activation of their degradation by site-specific proteases such as cathepsin D and calpains (for review, [5]), or by autophagosomes (linked to ATG modification; for review, [6]) and activation of the ubiquitin–proteasome system (UPS; linking multiubiquitination to proteolysis; for review, see [7]). Activation of the UPS is induced in stressed muscles by specific atrogenes that function as ubiquitin E3 ligases to multiubiquitinate muscle proteins, thereby subjecting them to proteasome-dependent degradation [8,9,10]. So far, two ubiquitin ligases, MAFBx (“muscle atrophy F-box protein”, also called atrogin-1) and MuRF1 (“muscle-specific RING finger protein-1”), and their relationship with muscle diseases, have been characterized in the most detail (for review, [11,12]). Both MAFBx and MuRF1 transcripts are markedly upregulated during muscle wasting states, thereby coupling enhanced muscle catabolism to UPS activation and promoting atrophy development by the downregulation of the IGF-1/Akt/mTOR pathway [13,14,15,16]. In addition to general muscle wasting, MAFBx and MuRF1 are upregulated by specific hormones and cytokines such as glucocorticoids [15], AT-II [17], and TNF-alpha [18], all factors known to promote muscle protein catabolism.

Consistent with their function as muscle atrogenes, inactivation of both the MAFBx and MuRF1 genes in murine models slow down muscle loss after nerve lesions [13,15]. Accordingly, the inhibition of cachexia promoting E3 ligases could be a potential treatment strategy to attenuate secondary myopathies. In line with this rationale, we recently identified MuRF1 inhibiting small molecules that slow down muscle mass and force loss in cardiac and cancer cachexia, respectively [19,20,21]. Here, we investigated if two compounds from this novel class of (2-oxo-chromen-7-yl)-heteromethyl-benzoic acids; i.e., MyoMed-205 and MyoMed-946 [21]) also have the potential to attenuate a secondary myopathy in a type 2 diabetes mellitus (T2DM) murine model. Since a previous study with MuRF1 overexpressing transgenic mice (“MuRF1 transgenes” or brief MuRF1-Tg) implicated MuRF1 also in glucose–insulin regulation [22], we first studied the effects of MuRF1 gene inactivation on glucose metabolism in mice. We included knockout mice for its close homolog MuRF2, because both E3 ligases target a highly overlapping set of proteins [23]. Data from the knockout (KO) models confirmed that both MuRF1 and MuRF2 participate in glucose and, also, in lipid regulation, a finding that should be considered when developing MuRF1/2 treatments. Next, we evaluated the effects of MyoMed-205 and MyoMed-946 in a diet-induced obese (DIO) T2DM mouse model on muscle functions; MyoMed-205 attenuated the loss of the holding impulse during progressing myopathy in DIO mice, therefore having potential in diabetes management.

## 2. Results

### 2.1. Glucose and Lipid Alterations in MuRF1-KO and MuRF2-KO Mice Implicate MuRF1 and MuRF2 in Carbohydrate and Fat Metabolism

As an initial survey for metabolic effects after MuRF1 and MuRF2 gene inactivation, we performed clinical chemistry on sera obtained from 16-h food-deprived MuRF1-KO, MuRF2-KO, and MuRF1-Tg mice, respectively. MuRF1-KO mice had elevated serum glucose and elevated triglycerides (Figure 1A,B). Additionally, consistent with a connection between MuRF1 and serum glucose, MuRF1-Tg mice were hypoglycemic (Figure 1C). MuRF2-KO mice also had elevated serum glucose (Figure 1A). In addition, electron-dense glycogen particles in musculus quadriceps (QUAD) tissues indicated a failure of MuRF2-KO mice to mobilize their glycogen stores during 16-h calorie deprivation (Figure 1D).

Next, we tested in both MuRF1- and in MuRF2-KO mice their tolerance to glucose by intraperitoneal glucose tolerance tests (IPGTTs) and to fat by fat-rich diets, respectively. IPGTT indicated a lowered glucose tolerance in MuRF2-KO mice at 30, 60, and 120 min after glucose injections (Figure 2A,B). In addition, MuRF2-KO mice when fed with a fat-rich diet became significantly more obese than wildtype (WT) mice after 13 weeks of dietary challenge (Figure 2C). In MuRF1-KO mice, we did not detect significant differences in serum glucose or body weight when compared to WT during IPGTTs or fat feeding, respectively. When determining the serum insulin, we noted a significance for elevated glucose/insulin ratios MuRF1-KO mice 30 min after glucose injection. (Figure 2B).

### 2.2. Effects of MuRF1- and MuRF2-Inactivation on Akt, Foxo3a, SDH, Complex-1, and PDC Activities

Next, we studied the glycolysis signaling pathway members in murine myocardium, QUAD, and tibialis anterior (TA) tissues in search for potential connections between MuRF1 and MuRF2 and glucose regulation. Western blot studies using Akt- and Akt-phosphoserine-473-directed antibodies indicated that phospho-serine 473 Akt was markedly elevated in MuRF1 and in MURF2-KO myocardial, QUAD, and TA extracts, while the total Akt remained normal (Figure 3A–D). This finding was also noted in cultured cardiomyocytes established from MuRF1- and MuRF2-KO hearts (Figure 3B), suggesting that the connection between MuRF1, MuRF2 and Akt is cell-autonomous. In MuRF1-KO extracts from the tibialis anterior (TA) muscle, phospho-FOXO3a was also elevated (Figure 3D). Interestingly, TA from MuRF2-KO mice conversely had lower levels of phospho-Foxo3a (Figure 3D). Finally, we found that MuRF1-KO myocardial extracts had significantly lower succinate dehydrogenase (SDH) and mitochondrial oxidative complex-1 (complex-1) activities than WT (Figure 4A,B). Additionally, consistent with MuRF1-mediated effects on the mitochondrial metabolism, we found that the re-expression of MuRF1 by AAV9 (adeno-associated virus, substrain 9) significantly elevated pyruvate dehydrogenase complex (PDC) activity when determined as described by Jeoung et al. [25], (Figure 4C). Previous yeast two-hybrid screens identified the PDC inhibitor PDK2 (pyruvate dehydrogenase kinase, isoform 2) as a potential binding partner of both MuRF1 and MuRF2 [23]. Therefore, we tested if PDK2 is a potential substrate of the MuRF1 and MuRF2 E3 ligases. The results indicated that MuRF1 and MuRF2 catalyze the mono-ubiquitination of PDK2 in vitro (Figure 4D). Therefore, possibly, MURF1 and MuRF2 can relieve PDC from PDK2 suppression. In conclusion, the regulation of phospho-Akt-Ser473, SDH, complex-1, and PDC are implicated by our MuRF1 and MuRF2 studies.

### 2.3. Adenoviral Transfer of MuRF1 into Myocardium Rescues KO Mice from Hyperglycemia

Since the KO models used in this study are not under Cre-inducible control, MURF1 and MuRF2 are inactivated throughout the entire development and in all tissues. Therefore, we tested next if an adenoviral reintroduction of MuRF1 could reverse the metabolic alterations noted above. The intravenous injection of recombinant AAV9-MLC2-MuRF1 virus stocks established the stable long-term expression of MuRF1 in the myocardium (Figure 5A). AAV9-MLC2-MuRF1 injections normalized the serum glucose in MuRF1 mice (Figure 5B). With regards to triglycerides, we also noted a trend to normalization (Figure 5C). As noted above, the injection of AAV9-MLC2-MuRF1 elevated the PDC activities in myocardium (Figure 4C).

### 2.4. Treatment of DIO Mice with MyoMed-205 Improves Holding Impulses in WHTs and Normalizes the Lymphocyte–Granulocyte Differential Blood Counts

Next, we studied the effects of the recently described small molecules that inhibit MuRF1 [19,20,21] in a mouse model for T2DM. We hypothesized that such a treatment might attenuate the myopathy accompanying T2DM by inhibiting the MuRF1 atrogene functions. Based on the KO data, diabetes should be monitored in such a treatment as well. As a T2DM disease model, we selected DIO mice [26]. As small molecules, we selected MyoMed-205 and MyoMed-946, which we recently found to protect the muscles during melanoma tumor-induced cancer cachexia [21]. Their feeding to DIO and WT mice over 28 days was well-tolerated. The mice behaved normally during the 28-day treatment period. Additionally, the tested small molecules of MyoMed-946 and MyoMed-205 did not alter the basal glucose or glucose tolerance significantly (Appendix A). The differential blood counts responded favorably to the compound feeding. DIO mice responded to diabetic stress with lymphopenia (Figure 6). In the compound-fed DIO groups, the lymphocyte–granulocyte ratio and absolute levels became normalized (Figure 6). Compound-feeding with MyoMed-205 also rescued the muscle holding impulse loss, as detected by WHTs from days 3 to 28. The progressive loss of the holding impulse in DIO mice was attenuated in the MyoMed-205-fed mice, while MyoMed-946 here was not effective (Figure 7).

## 3. Discussion

Previous works on MuRF1 focused on its roles as an atrogene [8,13,14]. However, its expression in healthy muscle tissues and impacts in other adaptive processes such as regeneration suggest a broader role of this E3 ligase in homeostasis. Recent works have also identified the titin filament and its binding partner Tcap/telethonin regulated by the mammalian clock gene in response to the circadian rhythm [27]. Tcap and titin, in turn, are targeted by MuRF1 [28,29]. Finally, the regulation of MuRF1 by branched chain amino acids (BCAAs) [30] supports the idea that MuRF1 integrates numerous metabolic and activity inputs for fine-tuning the dynamic turnover of myofibrils so that amino acids from the myofibril pool will be provided in the bloodstream under stress states such as starvation (see Appendix A). Conversely, the suppression of MuRF1 by BCAAs—in particular, leucine—appears to favor the synthesis of new sarcomeres when sufficient amino acid building blocks are available in the diet [31,32].

To investigate the MuRF1 roles in physiological energy regulation, we first determined the clinical chemistry after 16-h starvation. MuRF1-KO mice showed hyperglycemia, hyperlipidemia (Figure 4D), and a reduced glucose tolerance at 30 min (Figure 2). We also included MuRF2 in our studies, because MuRF1 and MuRF2 appear to share significantly overlapping signaling roles [23]. MuRF2-KO mice were also hyperglycemic, showing a more expanded reduction of glucose tolerance and, in addition, an intolerance to a fat-rich diet (Figure 2B,C). In order to find the possible underlying mechanisms, we analyzed the Foxo-Akt pathway. While MuRF1 is regulated by the Foxo3-Akt-mTOR pathway, which is related to protein synthesis [33,34], Foxo3-Akt also connects to carbohydrate metabolism [35,36], including the action of insulin on Akt by inducing phosphorylation at serine-473, a central step for the intracellular insulin signaling cascade [37]. When monitoring Foxo, Akt, and their phospho-activated forms in MuRF1- and MuRF2-KO mice, we found markedly upregulated Ser-253 phosphorylated Foxo3a and Ser-473 phosphorylated Akt in MURF1-KO the TA muscle (Figure 3). Phospho-473 Akt was also elevated in cultured primary cardiac myocytes obtained from the MuRF1- and MuRF2-KO myocardium, respectively (Figure 3B). Therefore, the suppression of Akt activation by MuRF1 and 2 appears to be myocyte cell-autonomous and not to require insulin. Interestingly, MuRF2- and MuRF1-KO mice responded differently with regards to Foxo3a phosphorylation, as MuRF2-KO augmented phospho-Ser-253 Foxo3a (Figure 3D and Appendix A).

Next, we hypothesized that a blockade downstream of Akt might cause features resembling a T2DM-like syndrome in MuRF1- or MuRF2-KO mice, despite a continuously nonphysiologically augmented Akt activation (similar perhaps to a reported T2DM model where diabetes developed after insulin/lipid infusion by metabolic feedback in the presence of intact IRS-1, Akt, and AS160 signaling, as described by Hoy et al. [38]). Therefore, we tested the PDC, SDH, and complex-1 activities, i.e., enzymes that, downstream of the cytosolic glycolytic pathway, catalyze the conversion of pyruvate to acetyl-CoA (coenzyme A) and its oxidation in the mitochondrial membrane (for metabolism regulation by PDC, see Sugden et al. [39]). The SDH and complex-1 activities were significantly downregulated in the myocardium in MuRF1-KO mice. A possible mechanism might be that MuRF1 may negatively regulate PDK2 by its E3 ligase activity (Figure 4D). Energy depletion by starvation that upregulates MuRF1 (Appendix A) then might relieve PDC from PDK2 suppression, thus augmenting the influx of acetyl-CoA in key organs such as the myocardium. Consistent with this model, AAV9-MLC2-MuRF1 expression increased the PDC activity in M1KO mouse hearts (Figure 4C).

The therapeutic targeting of MuRF1 may involve viral approaches or small molecules, the latter having, so far, more clinical applications. With regards to AAV approaches, a single injection of AAV9-MuRF1 led to a stable myocardial detectable expression for the nine months monitored in our study. We originally designed an MLC2-driven construct in an attempt to design an antihypertrophic intervention, as MuRF1-KOs are highly susceptible to cardiac hypertrophy, as described by Willis et al. [40]. However, using in vivo imaging of AAV9-treated MuRF1-KO and WT mice, we could not detect an effect on the cardiac trophicity (data not shown). Instead, and surprising to us, we noted a normalization of the serum glucose and a partial rescue of PDC. Follow-up studies are required to show how a cardiac-specific expression of a muscle-specific gene, such as MuRF1, by AAV9-MLC2 can affect the serum glucose, as this implies novel pathways or off-target effects outside the myocardium.

Finally, we studied the recently described MuRF1-inhibiting small molecules in DIO mice, because we wanted to determine if the muscle-protective effects from “atrogene” inhibition or potential pro-diabetic effects dominate in T2DM in this context. The treatment of DIO mice with MyoMed-205 and MyoMed-946 for 28 days did not cause significant effects on the basal glucose (Appendix A). When analyzing the blood of treated DIO mice further, we noted significant lymphopenia and granolocytosis. Lymphopenia combined with granolocytosis is found in social [41] or post-surgery stress states [42]. More recently, lymphopenia has been correlated with a significantly increased mortality in a large human cohort study [43]. Small-molecule compound-treated mice at day 28 had normalized lymphocyte–granulocyte counts (Figure 6). When monitoring the muscle functions by WHTs, the compound MyoMed-205 was also effective in attenuating the holding impulse loss. A better efficacy towards MuRF1 of Myomed-205 when compared to MyoMed-946 might be due to its higher bioavailability, because medicinal chemistry modifications attenuated the degradation of MyoMed-205 by serum esterases. In line with this hypothesis, in our recent murine melanoma cancer cachexia intervention study, MyoMed-205 was also superior over MyoMed-946 with regards to protecting the muscles from wasting and, also, in WHTs, as described by Adams et al. [21]. Future studies are warranted to test in more detail the differences between individual compounds and, also, their mechanistic actions.

In summary, our data on MuRF1- and MuRF2-KO mice suggest that both E3 ligases share roles in glucose metabolism regulation. Consistent with the intimate and redundant relationships between MuRF1 and MuRF2, both E3 ligases share highly similar interactomes, as described by Witt et al. [23]. Future studies will need to address how MuRF1 and MuRF2 cooperate in physiological stress signaling. While downstream functions such as glucose regulation are shared, different upstream stress signals may activate MuRF1 and MuRF2, respectively. For example, 48-h starvation induces MuRF1, but not MuRF2 (Appendix A), whereas the gene inactivation of MuRF1 and MuRF2 have opposite effects on FoxO3a phosphorylation (Figure 3D). Therefore, we hypothesize that myocytes under stress activate MuRF1 and MuRF2 differently, which, once in place, mediate the overlapping shared downstream adaptive responses during metabolic stress in turn.

## 4. Materials and Methods

### 4.1. Mouse Models and Phenotyping Studies

KO models for MuRF1 and MuRF2 were obtained as previously described by Witt et al. [23] and back-crossed with C57/Bl6 strain (Jackson Laboratory, Bar Harbour, Maine, USA) for >10 generations. WT (wild-type), MURF1, or MuRF2-KO male siblings were selected after genotyping and assigned to the study cohorts. Transgenic mice expressing MuRF1 under control of the muscle-specific creatine kinase promoter (MuRF1-Tg) mice were described by Hirner et al. [22]. MuRF1-Tg mice were backcrossed on a FVBN background for 8 generations and compared with their WT FVBN siblings (for FVBN mouse strain, commonly used for transgenic mouse production, see [24]). For clinical chemistry and before sacrifice, mice were deprived of food for 16 h but had access to water (5 p.m. to 9 a.m.). Murine musculus quadriceps (m. quadriceps or, in brief, QUAD), musculus tibialis anterior (TA), and myocardial tissues were dissected at sacrifice under a Zeiss biocular microscope (Zeiss GmbH, Jena, Germany). IPGTTs (intraperitoneal glucose tolerance tests) were performed as described by Hirner et al. [22]; blood was obtained from retrobulbar punctation and processed immediately. Fat-enriched food was obtained from “ssniff Specialdiäten” (Sniff GmbH, Soest, Germany, product D12492 high-fat diet (HFD) food with 60% calories from bovine fat) and supplemented with 1 g/kg compound for each respective treatment group. DIO-ICR mice (“DIO” groups) were used as a type 2 diabetes mellitus disease model and were obtained at Bienta (Enamine-Bienta, Kiev, Ukraine) by conditioning C57/Bl6 with a fat- and fructose-rich diet for 4 months, as previously described by Guan et al. [26]. After the conditioning phase, when DIO mice were 22% heavier than controls, eight DIO mice each were randomly assigned at the study begin to groups either continuing to receive the regular diet for an additional 16 days (DIO group) or a diet supplemented with 1 g/kg MyoMed-205 (DIO-205) or with MyoMed-946 (DIO-946). At day 17, mice were changed to low-calorie diets containing 50% less calories to challenge mice catabolically and to lose about 8% weight until sacrifice at day 28. Again, compounds were included in the low-calorie diets at 1 g/kg. As the control group (Con), ten nonconditioned C57/Bl6 mice were included, receiving the low-calorie diet without compounds.

Clinical chemistry for glucose and lipids was from whole fresh blood and determined at the Mannheim Clinic Clinical Chemistry Core; serum insulin was determined with the Crystal Chem Ultrasensitive Insulin Kit (catalog#90080, Crystal Chem Europe, Zaandam, Netherlands), as described by the manufacturer. Differential blood counts in DIO mice were determined by Bienta (Enamine-Bienta, Kiev, Ukraine).

Feeding with MyoMed-205 and MyoMed-946-spiked diets at 1 g compounds/kg rodent food was performed as recently described by Adams et al. [21]. Clinical chemistry was from whole fresh blood for all parameters. To assess the overall muscle function, wire hang tests (WHTs) were performed by a standard wire hang construction with a 40-cm-long wire and with a diameter of 2.5 mm placed at a height of 70 cm above the floor. Under the center of the wire, a large box with sawdust was placed. For testing, the mouse was hung to the wire with two limbs, and hang time was recorded. Each animal had three attempts for 180 s maximally each. After the three attempts, the maximal holding impulse was calculated (hanging time × body weight). All experiments on live mice followed the institutional animal care regulations and the approved protocols (MuRF1 and 2 KO and MuRF1-TG mice: Regierungspräsidium Karlsruhe: AZ 35-9185.81/G-108/07 and DIO mice at Bienta in Kiev: The Ukrainian Ministry of Education and Science (Ukrainian Ministry of Education of Science issued on 28-02-2019, registration number #1/11-2041).

### 4.2. Western Blot (WB) Studies

Anti-MuRF1 and MuRF2 antibodies were used as described previously by Witt et al. and Hirner et al. [22,23]. Akt, phospho Akt (Ser-473), FoxO3a, and phospho-FoxO3a (Ser-235) were commercially obtained: anti-Akt1 (pan; product #10176-2-AP), and anti-Phospho-Akt-Ser473 (product #664441-1) were from Proteintech (Proteintech Europe, Manchester, United Kingdom). Anti-Foxo3a (product PB9196) was from Boster (Pleasanton, CA, USA), and anti-Foxo3a-Phospho-Ser253 was from Abcam (product ab#154786, Cambridge, United Kingdom), respectively. For WB analysis, tissue lysates were prepared from snap-frozen tissues by powdering in a mortar. The homogenized powder was solubilized in 6-M urea buffer and separated on 4–10% PAGE essentially as described previously [19,20,21]. Blotted proteins were detected with alkaline phosphatase and quantified with AIDA software (Advanced Image Data Analyser, Version 2.11, for download see https://aidaimaging.com/download/), as described by Witt et al. and Hirner et al. [22,23].

### 4.3. SDH, complex-1, and Pyruvate Dehydrogenase Complex (PDC) Activity Measurements

Activities of mitochondrial enzymes involved in oxidative phosphorylation were determined from freshly prepared left ventricle myocardial extracts [44]. SDH and mitochondrial complex-1 activities were prepared as described [19,20]. Pyruvate dehydrogenase complex activity (PDCa) was measured by coupling the production of acetyl-coenzyme A to the acetylation of 4-aminoazobenzene-4′-sulfonic acid using recombinant avian arylamine *N*-acetyltransferase, as described by Jeoung et al. [25]. Change of absorption at 330 nm (SDH), 340 nm (complex1), and 460 nm (PDC) was measured on a Synergy 2 plate reader (Biotek, Bad Friedrichshall, Germany).

### 4.4. In Vitro Ubiquitination Assays

MuRF1 and MuRF2 full-length human cDNAs [45] were inserted into pETM-44 and expressed in *Escherichia coli* as hexahistidine maltose-binding protein (MBP) fusions. Shortly before setting up the ubiquitination reactions, the MBP tag was removed from MuRF1 and MuRF2 by proteolytic cleavage with precision protease (obtained from the European Molecular Biology Laboratory protein expression core facility). Human PDK2 full-length cDNA [46] was inserted into pETM-11 and expressed in *E.coli* as a hexahistidine-tagged fusion protein. Ubiquitination reactions were started by the addition of 75-nM E1, 1-µM E2 (UbcH5c), 150-µM ubiquitin, 4-µM ATP in 20-mM Tris/Cl, pH 7.5, 20-mM KCl, 5-mM MgCl2, 1-mM DTT, and 1-mM PMSF (phenylmethylsulfonylfluorid). E1, E2, and ubiquitin were purchased from BostonBiochem (Cambridge, MA, USA). Where indicated, PDK2 (2 µM), MuRF1 (1µM), or MuRF2 (1µM) were also added. Reactions were incubated for 1 h at 37 °C and then stopped by the addition of SDS-loading buffer. Samples were then analyzed by SDS-PAGE and blotted onto nitrocellulose. Specific products were detected with a rabbit polyclonal antiserum against PDK2 (Abcam, Cambridge, UK, product 68164).

### 4.5. Tissue Culturing

Primary cardiac myocytes were obtained from newborn MuRF1 or MuRF2 KO mice using a kit and as described in the provider’s manual (Pierce, Waltham, MA, USA, product 88281).

### 4.6. AAV9-Mediated Gene Transfer of MuRF1

cDNAs coding for full-length human MuRF1 were amplified with the below specific primers and inserted with BglII + Bsrg 1 into pds-AAV-CMVenh-MCL260 for the production of adenoviral-associated virus (AAV) particles, as described by Mueller et al. [47]. AAV subtype 9 (AAV9) virus particle production was performed essentially as described by Goehringer et al. previously [48]. For AAV9-mediated MuRF1 gene transfer, 2.5 × 10^11^ AAV9 virus particles in 50–100-µL phosphate-buffered saline (PBS) were injected into the tail veins of 10-week-old KO mice. The following primers were used (small letters denote restriction site tags):

MuRF1-S: tttagatct-AGGAGGCAGCTAGGCGTGGCTCTC and

MuRF1-R; ttttgtaca-TCTGGGGGCCTCTCATTCATCCAGCTC.

### 4.7. Statistical Analysis

Data are presented as mean ± SEM. One-way analysis of variance (ANOVA) followed by Bonferroni’s post hoc was used to compare groups, while two-way repeated measures ANOVA followed by Bonferroni’s post hoc was used to assess the contractile function (GraphPad Prism, downloaded from www.graphpad.com). Significance was accepted as *p* < 0.05.

## 5. Conclusions

MuRF1 and MuRF2 play a role in the regulation of glycolysis and lipid metabolism. This should be considered when targeting these genes during myopathy interventions. Small molecules targeting MuRF1 such as MyoMed-205 may be suitable to attenuate the muscle strength loss in T2DM conditions. Future studies need to address how MyoMed-205 affects holding impulses, as the present study was limited to WHTs.

## Figures and Tables

**Figure 1 ijms-22-02225-f001:**
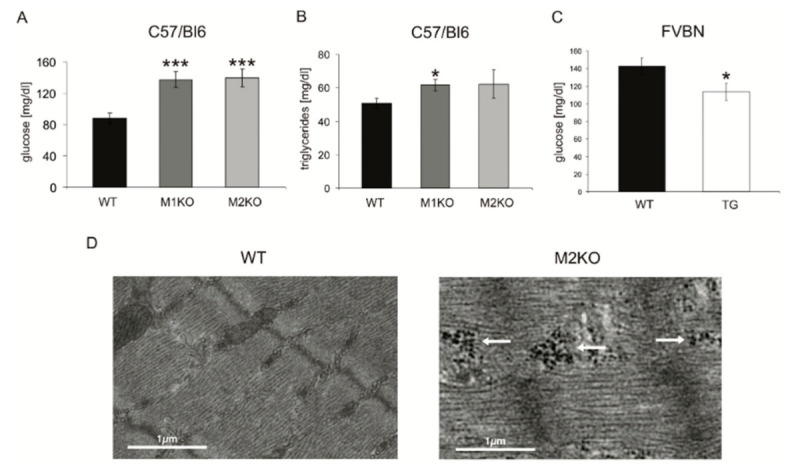
MuRF1- and MuRF2-KO (knockout) mice have elevated serum glucose and triglycerides. (**A**–**C**) Clinical chemistry of sera from MuRF1-KO (M1KO), MuRF2-KO (M2KO) and MuRF1-Tg (TG or trangene) mice. (**A**) MuRF1-KO and MuRF2-KO mice have upregulated serum glucose (*** *p* < 0.001; *n* = 20). (**B**) Triglycerides were elevated in MuRF1-KO mice (* *p* < 0.05, *n* = 20), and a trend was noted in MuRF2-KO mice (*p* = 0.1, *n* = 20). (**C**) Transgenic (“TG”) mice that overexpress MuRF1 in their skeletal muscles have lower serum glucose (FVBN mouse strain; see [24]. for details; * *p* < 0.05; *n* = 12). (**D**) Electron micrographs of the quadriceps (QUAD) from starved wild-type (WT) and MuRF2-KO mice. MuRF2-KO QUAD retains glycogen granules (white arrows) in its high molecular weight electron-dense macro-glycogen form after 16-h starvation (arrows) in contrast to WT.

**Figure 2 ijms-22-02225-f002:**
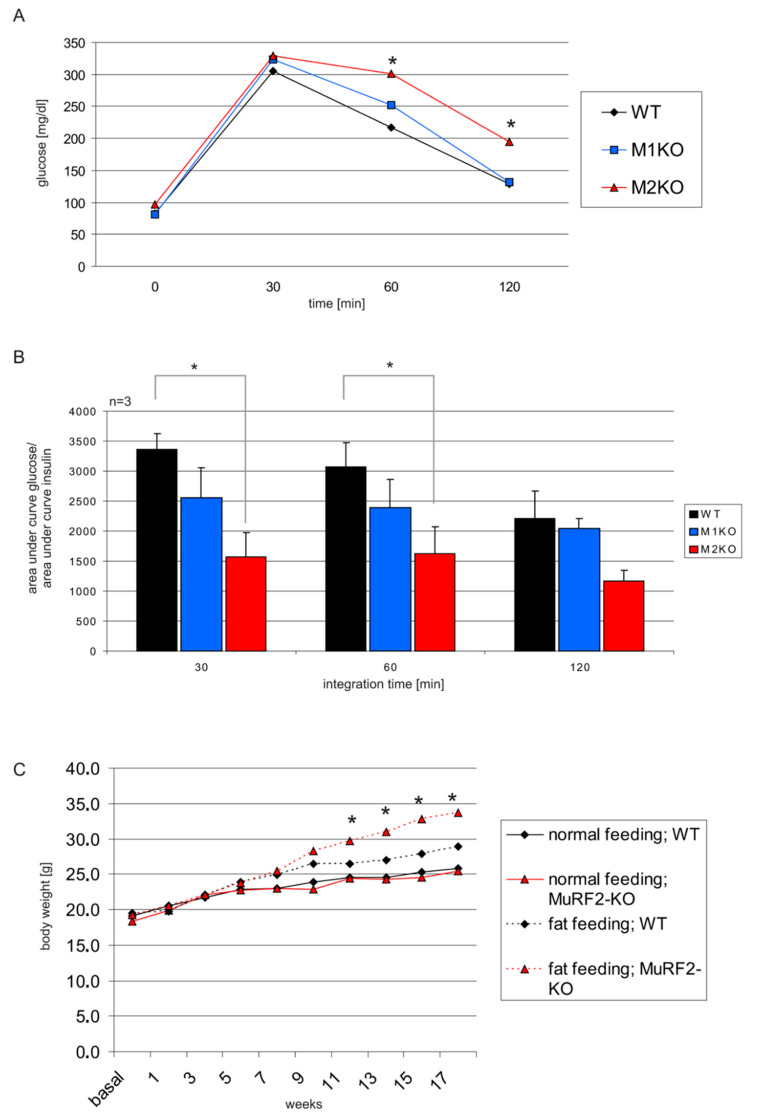
Reduced tolerance to glucose or fat in MuRF1 and MuRF2-KO mice. (**A**) Intraperitoneal glucose tolerance tests (IPGTT) on MuRF1- and MuRF2-KO mice. MuRF2-KO mice showed elevated glucose levels at the 60- and 120-min time points (* *p* < 0.05, *n* = 6). (**B**) Insulin was determined from blood samples from A, and the areas under the glucose and insulin curves were calculated. Respective ratios as an indicator for insulin sensitivity are shown. MuRF1-KO mice had, at 30 min, a reduced glucose–insulin ratio; MuRF2-KO mice have lowered glucose–insulin ratios at 30, 60, and 120 min, respectively (* *p* < 0.05, *n* = 6). (**C**) WT, MuRF1-KO-, and MuRF2-KO mice were fed with a fat-enriched diet. MuRF2-KO mice became progressively obese after 13 weeks when compared to WT (* *p* < 0.05, *n* = 6).

**Figure 3 ijms-22-02225-f003:**
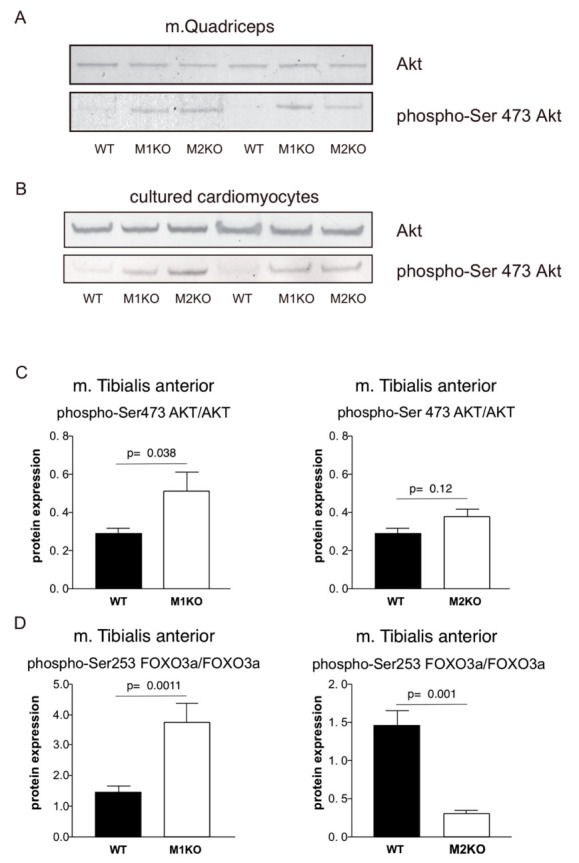
MuRF1 and MuRF2 inactivation perturbs Akt- and FOXO3a-phosphorylation. (**A**,**B**) Western blot screens with specific antibodies using 16-h-starved MuRF1- and MuRF2-deficient QUAD or cell extracts. (**A**) In QUAD, phospho-Ser-473 is upregulated, and total Akt is normal. (**B**) Augmented Akt-Ser-473 phosphorylation in cultured primary cardiac myocytes obtained from MuRF1 or MuRF2 fetal mice indicates the cell-autonomous MuRF1,2 dependent regulation of Akt. (**C**,**D**) Quantitation of Akt, Foxo, and their Ser-473 and Ser-253 phosphoforms, respectively, by Western blots from TA mouse muscle extracts. MuRF1 inactivation leads to elevated phospho-Akt and phospho-Foxo3a, respectively; MuRF2 inactivation instead lowered phospho-Foxo3a (MuRF1-KO, *n* = 6 and MuRF2-KO, *n* = 8; see Appendix A).

**Figure 4 ijms-22-02225-f004:**
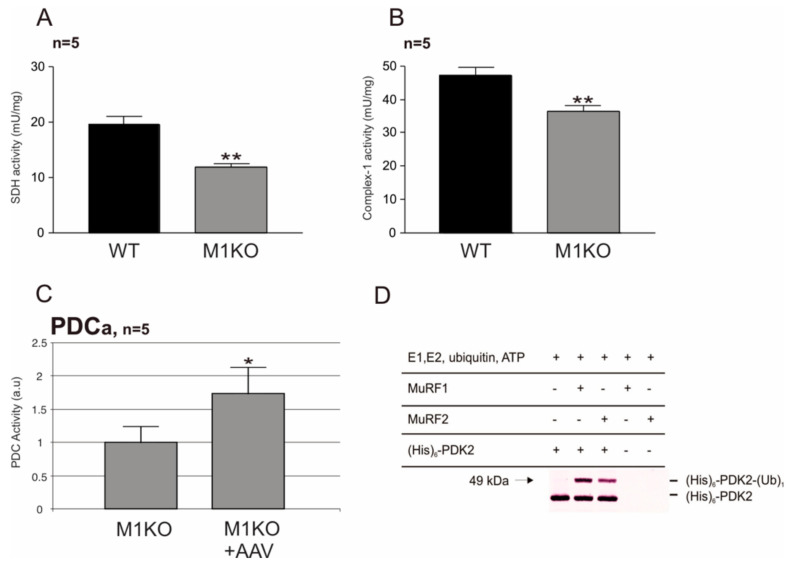
Effects of MuRF1 on SDH, complex-1, and PDC, and of MuRF1 and MuRF2 on PDK2. (**A**,**B**) SDH and complex-1 activities were measured in myocardial extracts by enzymatic assays [20,25]. MuRF1-KO myocardium had significantly lower SDH and complex-1 activities (** *p* < 0.01; *n* = 5). (**C**) PDC activities (PDCa) in myocardial extracts were measured by determining the conversion rate of pyruvate to acetyl-CoA (coenzyme A). PDCa is given in artificial units (AUs), where WT was defined as the reference as 1.0. AAV9-MCL2-MuRF1 (MLC2 indicates myosin light chain-2 promoter driven) injections elevated the PDCa (* *p* < 0.05, *n* = 5). (**D**) In vitro ubiquitination of PDK2 by MuRF1 and MuRF2. Addition of the full-length recombinant MuRF1 or MuRF2 E3 ligases to reaction mixes containing the PDC regulator PDK2 and the cofactors ubiquitin and E1 and E2 ligases results in the mono-ubiquitination of PDK2 as detected with specific antibodies (arrow).

**Figure 5 ijms-22-02225-f005:**
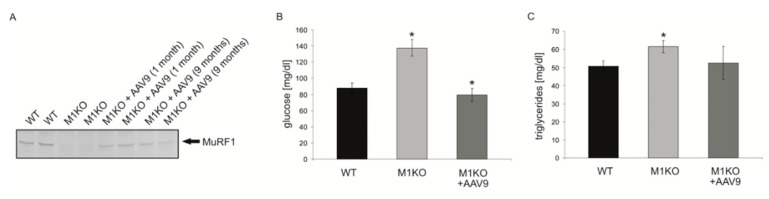
Rescue of metabolic alterations in MuRF1-KO mice after adenoviral re-expression of MuRF1 in the myocardium. (**A**) MuRF1 protein, physiologically expressed in the mouse myocardium from the left ventricle (WT lanes), is absent in MuRF1-KO mice (see, also, Appendix A). One and nine months after AAV9-MLC2-MuRF1 injection, respectively, MuRF1 is detected in the previously MuRF1-free myocardium. (**B**) Elevated serum glucose in MuRF1-KO mice is normalized after AAV9-MLC2-MuRF1 injections (* *p* < 0.05, *n* = 6). (**C**) Triglycerides were significantly elevated in MuRF1-KO mice when compared to the WT (* *p* < 0.05, *n* = 6). This significance is lost after AAV9-MLC2-MuRF1 injection.

**Figure 6 ijms-22-02225-f006:**
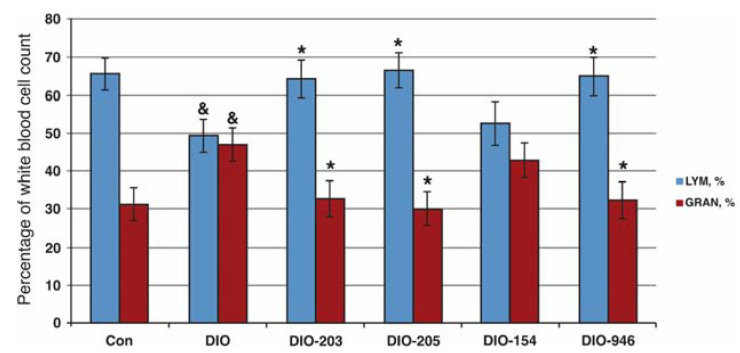
Normalization of lymphocyte and granulocyte counts during the compound treatments. Control mice had white blood cell counts with about 65% lymphocytes and 30% granulocytes (Con group). DIO mice, in contrast, had similar lymphocyte and granulozyte percentages at day 28 for *p* < 0.05, comparison of Con to DIO). The treatments with MyoMed-205 and MyoMed-946 normalized the white blood cell ratios. The treatments with MyoMed-203, MyoMed-205, and MyoMed-946 were all three closely structurally related and normalized the white blood cell ratios. In contrast, their metabolite MyoMed-154, which is generated by their hydrolysis in the serum, had no effect on the white blood cell counts (* *p* < 0.05 for the DIO+ compound compared to the DIO group, *n* = 8 and ^&^
*p* < 0.05 for DIO compared to the control group, *n* = 8).

**Figure 7 ijms-22-02225-f007:**
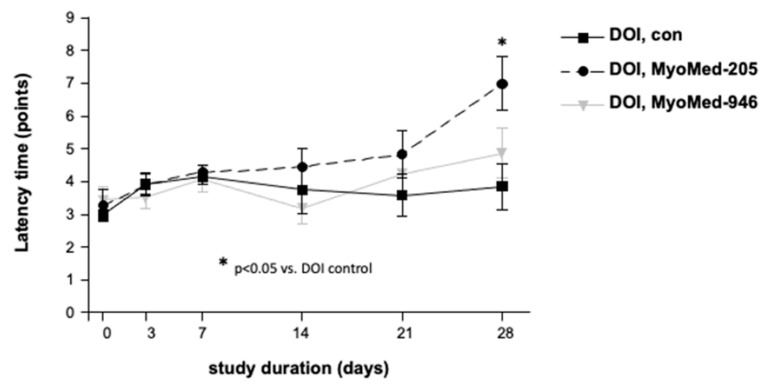
Protective effects of MyoMed-205 on the holding impulse in DIO mice. The muscle function was assessed by wire hang tests (WHTs) at days 3, 7, 14, 21, and 28. Mice in the DIO group have impaired holding impulses at day 28. Feeding with MyoMed-205 improved the holding impulse, whereas MyoMed-946 had no effect on the holding impulses in WHTs (* *p* < 0.05 DIO + 205 compared to the DIO group, *n* = 8).

## Data Availability

Not applicable.

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
