# Peer review of "Regulation of Glucose Metabolism by MuRF1 and Treatment of Myopathy in Diabetic Mice with Small Molecules Targeting MuRF1"

_ijms, 2021, doi:10.3390/ijms22042225_

Round 1

Reviewer 1 Report

General:

The authors study a highly important and interesting topic within the muscle research field, namely the contribution and precise function of the ligases MuFR1 and MuRF2. Using in vitro and in vivo approaches, the authors demonstrate that MuRF1 and MuRF2 knockout mice have altered glucose metabolism and related pathways. Finally, the authors used a small molecule inhibitor approach to silence MuRF1 and MuRF2 in a mouse model of type 2 diabetes mellitus (T2DM). Interestingly, the inhibitor Myomed-205 rescued muscle mass weakness in the T2DM model. The combination of the inhibitors Myomed-205 and Myomed-946 normalized immune responses in the T2DM model. These results are highly interesting and important as they demonstrate the importance of MuRF1 MuRF2 inhibitors as pharmacological agents to improve muscle-related weaknesses in a preclinical model.

Specific:

  1. Methods, line 87: 'Fettenergy' is a rather German word. This should be adapted to English terms.
  2. Methods, line 142: In general for this AAV9 section: Could the authors provide some additional information, e.g. primer sequences?
  3. Methods, line 148: Please provide information about the 'vector solution'.
  4. Figure 1D: Please provide a scale bar for the TEM pictures.
  5. Figure 3; The resolution of the figure is rather low. I assume this happened during the upload. Please provide higher resolution figures to allow the reader to evaluate the interesting results in clear detail.
  6. Figure 4: Rather small pictures, please provide higher resolution.
  7. Methods section: Please provide here also details on the source of the Foxo-antibodies, i.e. provider , catolog number which Foxo gene member.
  8. Discussion: Parts of the data suggest a close relationship of MuRF1 and MuRF2, whereas other data hint at differences in their regulation. Can the authors speculate on the relationship between MuRF1 and MuRF? In line with this: In their previous publication (ref 20), a knockdown of MuRF2 by compound was also noted. Therefore, can the authors expand the discussion by adding a novel short paragraph on how they see MuRF1 and MuRF2 roles when cooperating in myocyte signaling?
  9. Figure 6: In this experiment, in addition to Myomed-205 and Myomed-946, two additional small molecules were used. The reviewer understands that the authors can at this stage not yet disclose structural formula as their patent filing is pending. But can the authors at least provide one line in the legend how the two additional analogs relate to Myomed-205 and Myomed-946, (i.e. structural similar or novel lead compounds?).
  10. Figure 7: The improvement of holding impulse in DIO mice during Myomed-205 feeding is impressive. Do the authors believe that force loss is rescued only? Or is it possible that an increase occurs? In their recent publication, Myomed-205 appeared to increase muscle strength above controls in their 1RM test (ref 21).

Reviewer 2 Report

Although the manuscript was very scientifically sound, the manuscript preparation requires some significant modification before it should be considered publishable.  There are multiple minor errors throughout the text and there are some major discrepancies between the description of the results and the graphs that are being referenced.  Here are my suggested edits:

Major

For figure 2B, there is no mention of the insulin measurement in the actual results section. This makes it confusing to the reader when describing the results.

In the results section 3.2, there are many discrepancies between the results described in this section and the graphs that represent the data. This entire section and corresponding graphs needs to be carefully reviewed for accuracy before resubmitting. The discrepancies I found are as follows:

Lines 191: The results are describing only the findings for MuRF1, however, it appears that Phospho-FOXO/FOXO was decreased substantially in MuRF2, but this was unmentioned in the results.

All figures (A-D) for figure 3 are of very low image quality. The information on the graphs is very difficult to see. A higher image quality should be provided for these graphs

Line 202: The results section describe the information in this graph as tibialis extract, but the WB chart is labeled as cultured cardiomyocytes.

Line 204: Graphs C and D are described as cardiomyocytes, but the caption under the graphs describes them as TA extracts. 

Line 209: Graph 4A is described as QUAD tissue in section 3.2, but in the caption it is described as myocardial. 

Line 224: This line references Graph 5C as being a Glucose measurement, but it is a triglyceride measurement

Graphs 4A and 4B are also of very low image quality and should be improved

Figure 7 is also of fairly low image quality and is difficult to read

Minor

Line 51: The full names of MAFBx and MuRF1 are not previously listed prior to abbreviation. This is typically done during the first mention of these factors

Line 81: This is an incomplete sentence. It is unusual to end a sentence with just a reference number in parentheses when the sentence is otherwise incomplete. Typically, the author would state "are described by ____ et al." 

Line 83: I am assuming that the M.quadriceps stands for murine quadriceps. This needs to be stated before abbreviating

Line 101: This is the same issue as observed with Line 81. This should be corrected

Line 107: There is a typographical error. Life should be "live"

Line 114: There should be a comma after Danvers

Line 116: should the U in urea be capitalized. I may be mistaken, but I think this is incorrect. 

Line 124: This sentence should likely be ended with the word "previously"

Line 134: The company listed is called BostonBiochem. It is incorrectly listed. It is also not cited for location

Line 141: I believe that this is supposed to say Pierce manual, not "priced"

Line 160: The way that this sentence is currently worded, it sounds like MuRF1 is the only group getting the GTT, while MuRF2 is the only group getting the fat rich diet. This needs to be reworded

Figure 1A: The asterisks representing significance are fairly off-center. For manuscript homogeneity, this should be addressed

Line 250: The sentence is incomplete here. It appears to be simply missing words. 

Line 274: the "s" in BCAAs should be lower case. 

References: There are some major discrepancies within the formatting of the references. Some are numbered, other are not. The references that are numbered twice are off count as well. The reference section should be carefully reviewed for accuracy and consistency. 

Reviewer 3 Report

Siegfried et al reported that Thus, small molecules directed to MuRF1 may be useful to attenuate skeletal muscle dysfunction in T2DM conditions. Several critical issues need to be improved.

  1. The data presentation is poor. The first letter of the word in each figure should be capital letter.
  2. The function assay of muscle function is lacked. Such as strength.
  3. Resolution of figures are poor.
  4. Several critical controls are missed. For example, no loading control of fig 5a.
  5. N=3 in animal study is insufficient.

Round 2

Reviewer 3 Report

My questions had been addressed. This paper is acceptable.

Author Response

Thank you for your comment.